# On the Symbolic Use of Dragons by Jacobus de Voragine and J. R. R. Tolkien

Camilo Peralta 

English Department, Fort Hays State University, Hays, KS 67601, USA; cgperalta@fhsu.edu

**Abstract:** This article focuses on the symbolic use of dragons in several works by J. R. R. Tolkien and *The Golden Legend*, a popular compilation of saints' lives by Jacobus de Voragine. In the medieval tradition, as recounted by Voragine, dragons serve as symbols of powerful evil through which the inherent weakness of postlapsarian ("after the Fall") humans can be emphasized. The sudden, miraculous defeat of dragons also illustrates what is possible through faith and the grace of God, anticipating Tolkien's notion of *eucatastrophe*, the unexpected reversal in fortune that characterizes the best fairy tales, which is now recognized as a key component of his own approach to mythopoeia.

**Keywords:** dragons; J. R. R. Tolkien; Jacobus de Voragine; *The Golden Legend*; eucatastrophe



## 1. Introduction

Glaurung. Ancalagon the Black. Smaug. By now, most fans of J. R. R. Tolkien's legendarium are as familiar with the names of his great dragons as they are with those of the nine-member Fellowship. Tolkien's dragons are powerful and majestic creatures, all the more interesting for their relative scarcity in Middle-earth; even at the height of their glory in the First Age, most of them make only a brief appearance at the very end of the War of Wrath. Smaug serves as the primary antagonist of Tolkien's 1937 children's fantasy novel *The Hobbit*, but he and his kin play no major role in *The Lord of the Rings*, which was published between 1954–1955 and represents the most mature mythopoeic work completed by Tolkien during his lifetime. It may be that Tolkien simply could not figure out how to incorporate dragons into his story of the destruction of the One Ring, but in a 1937 letter written to his publisher, Stanley Unwin, he suggests another explanation:

> Mr. Baggins began as a comic tale among conventional and inconsistent Grimm's fairy-tale dwarves, and got drawn into the edge of it so that even Sauron the terrible peeped over the edge. And what more can hobbits do? They can be comic, but their comedy is suburban unless it is set against things more elemental. But the real fun about orcs and dragons (to my mind) was before their time.
>
> (Tolkien 2000, p. 26)

One reason that Tolkien did not include dragons in *Lord of the Rings*, then, is because he very sensibly realized that they would make an incongruous match for the hobbits who comprised the core of the Fellowship. The "real fun" of dragons, their appeal to him as an author, lay in the use he could make of them during the First and Second Ages, long before the time of the hobbits.

As Tolkien was undoubtedly aware, dragons underwent a similar process of diminishment in the real world. In "On Fairy-stories," he mentions never having believed in them as a child, arguing that "desirability" is a more important factor in the success of such tales than "possibility": "If they awakened desire, satisfying it while often whetting it unbearably, they succeeded" (Tolkien 1966, p. 63). However, for generations of children, reared on the rich dragonlore of pagan mythology and the literature of the Middle Ages, dragons must have seemed a very different matter. Having learned from these stories to regard them as real creatures, these children surely did not desire dragons so much as they

feared them! That terror is something Tolkien managed to capture in his own portrayal of them. In this paper, however, I am more interested in some of the other ways that the dragons of medieval literature might have inspired him, especially in regards to the key themes of postlapsarian weakness and eucatastrophe in *The Hobbit*, *The Lord of the Rings*, and *The Silmarillion*. I will focus on the parallel treatment of dragons in these texts and in the *Legenda Aurea* or *Golden Legend*, a collection of saints' lives compiled by Jacobus de Voragine in the thirteenth century. One of the most widely-read works of the Middle Ages, *The Golden Legend* almost certainly influenced the creation of Middle-earth: Judy Ann Ford (2011) describes it as one of the primary medieval resources used by Tolkien "to shape those elements of Middle-earth that seem magical to modern audiences" (p. 134). It contains several stories involving dragons that appertain more closely to the themes mentioned above than some of the texts known to have strongly influenced his conception of dragons, such as *Beowulf* or the *Völsunga saga*. At the same time, I am not as interested in how Jacobus' dragons may or may not have influenced their counterparts in Middle-earth, as I am in exploring the parallels between their authors' respective uses of them.

## 2. Postlapsarian Weakness in the Works of Jacobus de Voragine and Tolkien

Pre- and postlapsarian are terms used in Christianity to refer to the general condition of human beings before and after the Biblical Fall, respectively. As recounted in the second and third books of Genesis, God created Adam and Eve free from sin and in a state of "original justice," in which "[t]he inner harmony of the human person, the harmony between man and woman, and finally the harmony between the first couple and all creation was perfectly realized and maintained" (*Catechism of the Catholic Church* 1997, p. 376). They did not experience suffering or sorrow, but enjoyed a kind of blissfully nescient existence, full of "simplicitie and spotless innocence," in Milton's memorable description of life in the Garden of Eden (Milton [1667] 1971, p. IV.318). After being tempted by Satan into violating the one commandment given to them by God not to eat from the Tree of Knowledge of Good and Evil, Adam and Eve were punished by being deprived of their former state of original justice. Henceforth, they would have to labor "in sorrow" to produce either children or food (Gen. 2:16–17); because they no longer had access to the Tree of Life, they would eventually weaken from sickness or old age, and die. Their Original Sin was a doom that would be inherited by all of their descendants, ensuring that the entire human race was born in a similar state of "postlapsarian weakness," characterized by a permanent decline in their will, reason, and physical strength.

### 2.1. Textual Support

There is strong textual support in the Bible for this view, which, in addition, has been affirmed by major thinkers in the Catholic tradition to which Jacobus and Tolkien both belonged. The weakness of fallen humans is immediately apparent in the decreasing longevity of Adam's descendants. Adam, we are told, lived a total of 930 years, his son, Seth, 912, and his grandson, Enos, 905. Despite a few aberrations after Enos, there is a general trend downward, until we get to Lamech, the father of Noah, who lived only 730 years (Gen. 5:5–31). Whether one subscribes to a literal or allegorical interpretation of such verses, it is clear that Moses (by tradition regarded as the author of the Pentateuch or Torah, the first five books of the Bible) wanted to emphasize the continual decline in the average lifespan of humans since the Fall. In his *Epistle to the Romans*, Paul stresses some of the spiritual consequences of Adam's sin for all of creation, which was "made subject to vanity, not willingly," but by God as punishment for what he and Eve had done. He adds, "For now we know that the whole creation groaneth and travaileth in pain together until now" (Rom. 8:18–20). Only by embracing God's grace as extended to us through His son's sacrifice on the cross can humans hope to be saved from the inherent disorder and weakness of our souls (2 Cor. 12:9). Likewise, Augustine (2008) describes God as "the Author of all natures but not of their defects," explaining that He had "created man good; but man, corrupt by choice and condemned by justice, has produced a progeny that is

both corrupt and condemned" (XIII.xiv, pp. 316–17). As with the dwindling of human lifespans since the Fall, there has long been a tendency to view the defects, corruption, and condemnation mentioned by Augustine here in a progressive sense: i.e., as problems that are growing worse over time.

### 2.2. Mythic and Artistic Depiction of Dragons

Any discussion of dragons must begin by acknowledging the long, complicated development of the dragon tradition in the West. Etymologically, the word is derived from Greek (*drakon*) through Latin (*draco*) and Old French (*dragon*); these cognates all refer to a "large serpent" or sea monster (Harper 2023). Greek mythology features a number of snake or worm-like creatures that were sometimes referred to as or associated with dragons, included the dreaded Amphisbaena, described by Pliny the Elder as a snake with heads on both sides of its body (Syropoulos 2018, p. 41). Hesiod's *Theogony*, which dates to the eight or seventh century B.C., mentions a cataclysmic battle between Zeus and the serpent Typhon, who resembles a multi-headed dragon and is often credited as the father of several important dragons (Hesiod 1993, pp. 826–85). The Bible provides evidence of additional influence on the dragon tradition from Mesopotamia and South Asia. The Hebrew word *tannīn* refers to a "serpentine sea creature" mentioned throughout the Old Testament (Yoder 2013, p. 488), which the translators of the King James Version of the Bible chose to render variously as "great whale" (Gen. 1:21) or "dragon" (Isa. 27:1). In sum, it is important to remember that the now-familiar image of the dragon from modern fantasy is only a recent development. We often use the term to describe creatures that medieval writers such as Jacobus de Voragine would have known by a variety of names. In *The Golden Legend*, many of the dragons are of the flightless, worm-like kind, though I believe they can be still referred to, accurately, as dragons, especially since Tolkien's earliest dragons (e.g., Glaurung) also lack the ability to fly.

It may also be helpful to explore some of the dragon iconography produced during the classical and medieval periods, since these visual representations of dragons often proved as influential—if not more so—than the textual ones. Sharon Khalife-Gueta, for example, traces the close association of serpents, dragons, and goddesses through ancient Levantine, Egyptian, Minoan, and Greek art. She points to the image of Medea "in a dragons-driven chariot surrounded by the cycle of the fiery sun" on a fifth century BC vase as a source of inspiration for depictions of Daenerys Targaryen on the popular HBO series *Game of Thrones*, which illustrates the powerful influence of dragon iconography through the centuries (Khalife-Gueta 2022, p. 85). Many visual portrayals of dragons provide clear evidence of the real or fanciful creatures on which they were based and reveal elements that would eventually be incorporated into the literary traditions about them. For example, in Mesopotamian art, dragons often incorporate characteristics of the deity known as Anzû or Imdugud, who was usually depicted as either a fire-breathing bird or a lion-headed eagle. It is clear from these images that their role as *protectores o guardianes de tesoros*, "protectors or guardians of treasure," so familiar to us from *Beowulf*, the *Völsunga saga*, and Tolkien's own depiction of dragons, had already been firmly established by the start of the Middle Ages (Cuadra 2012, p. 111). Finally, illustrations in late medieval copies of the eighth century *Commentary on the Apocalypse* by Beatus de Liébana reveal how the appearance of dragons was often conflated with that of other beasts, such as lions, griffins, and giant worms in Christian art. These manuscripts also support the Biblical tradition of identifying dragons as representatives or symbols of Satan (Consiglieri 2016, p. 90).

### 2.3. Dragons and Postlapsarian Weakness

What does all of this have to do with either the Middle Ages or Middle-earth? As devout Catholics, Jacobus and Tolkien were certainly familiar with Christian teachings about Original Sin and the portrayal of dragons in art and the Bible. Both tend to follow the examples of the prophet Isaiah and John the Evangelist (author of the Book of Revelations), who present dragons as powerful, evil beings that cannot easily be combatted, let alone

overcome, by ordinary humans (Isa. 27:1, Rev. 20:2). The encounter between St. Martha and a dragon is indicative of how such incidents are treated in *The Golden Legend*. It begins with a detailed description of the monster: "half animal and half fish, larger than an ox, longer than a horse, with teeth like swords and as sharp as horns, and two bucklers on each side of his body" (de Voragine 1948, p. 392). This dragon is only slain through the combined efforts of an entire village after Martha succeeds in binding him with her own, blessed girdle. St. Silvester's dragon is even deadlier: "Holy Emperor," the people complain to Constantine, "there is a dragon in a cave, and since thou didst receive the faith of Christ, this dragon daily slays more than three hundred men with his breath!" (de Voragine 1948, p. 70). Of all the stories involving saints and dragons recounted by Jacobus, only St. George's features an actual physical battle between man and beast. However, since it ends in a miraculous (or eucatastrophic) manner, as I shall discuss below, it can be safely dismissed as an exception that proves the rule. For the most part, Jacobus does not depart from the general treatment of dragons in the Bible, which depicts them as powerful symbols of evil opposed only by God or His chief ministers such as Michael the Archangel. Of course, it is worth recalling that he did not invent most of the legends he recounts, but compiled them from a variety of sources (Mula 2003, p. 178). Rather than attribute the depiction of dragons in *The Golden Legend* to a single author, therefore, it may be more accurate to speak of their general treatment within the Christian tradition.

In any case, something like a postlapsarian decline is clearly evident in most of the races of Arda, Tolkien's invented world, including the nigh-immortal elves. "In the beginning," he writes in *The Silmarillion*, "the Elder Children of Ilúvatar were stronger and greater than they have since become . . . " (Tolkien 1977, p. 49). He does not elaborate on the exact nature of their decline, but further distinguishes between two kinds of elves: the Calaquendi, who have seen the light of the Two Trees, and the Moriquendi, who have not. Among the former are the Vanyar, who were the first to reach Valinor and remained ever after the "beloved of Manwë and Varda," surpassing in fairness and nobility their late-arriving kin (Tolkien 1977, p. 53). There is a clear hierarchy of elves based on the duration and strength of their relationship with Eru and the Valar, which parallels that of human beings and God. There are no scenes of temptation set in the gardens of Middle-earth, but the elves experience repeated falls: the first "comes about through the possessive attitude of Fëanor and his seven sons" to the Silmaril, Tolkien explains in a 1951 letter to Milton Waldman, and the second occurs during the Second Age, after several elves decline the summons to return to Valinor and decide to linger in Middle-earth instead. "They thus became obsessed with 'fading,' the mode in which the changes of time . . . was perceived by them. They became sad, and their art (shall we say) antiquarian, and their efforts all really a kind of embalming" (Tolkien 2000, pp. 148–51). Long before the dawn of the Third Age, during which mankind first began to grow in power and influence, the elves' gradual diminution had already been established as a major theme, and one that anticipates humanity's own, inevitable downfall.

### 2.4. Elves and Humans

Indeed, the postlapsarian nature of Tolkien's world is further highlighted by some of the natural differences between elves and humans, Eru's eldest children and his youngest. "While Men are similar to the Sindar [a group of Moriquendi] in some respects," Martin Simonson (2017) observes, "they are pictured as a much lesser species" (p. 381). "Immortal were the Elves," Tolkien writes, "and their wisdom waxed from age to age, and no sickness nor pestilence brought death to them." By contrast, "Men were more frail, more easily slain by weapons or mischance, and less easily healed; subject to sickness and many ills; and they grew old and died" (Tolkien 1977, p. 104). Is it any wonder, then, that dragons feature so little in the stories of the Third Age, the Age of Men? They are capable of laying waste to entire armies of elves, the most powerful race in Middle-earth, and would easily overwhelm their weaker counterparts, including humans. Surely it is no coincidence that the single largest gathering of dragons in battle occurs during the War of Wrath at the end of

the First Age, when, facing imminent defeat, Morgoth unleashes his "last desperate assault" against the elves and Valar marching against him: "and out of the pits of Angband there issued the winged dragons, that had not before been seen; and so sudden and ruinous was the onset of that dreadful fleet that the host of the Valinor was driven back" (Tolkien 1977, p. 252). This spectacular assault force is unlike anything humans will ever face, a fact that can be regarded as a tacit admission of their physical and mental weakness in comparison to elves. However great the courage possessed by the greatest human dragon-slayers, Túrin Turambar and Bard the Bowman, they are, after all, only tasked with defeating a single dragon. Further, like the saints of *The Golden Legend*, their victories over Glaurung and Smaug depend not or not only on their own skill or luck, but on something close to a miracle.

### 3. Tolkien and Eucatastrophe

I refer here, of course, to eucatastrophe, "the sudden happy turn in a story which pierces you with a joy that brings tears" (Tolkien 2000, p. 100). In all of the best legends, myths, and fairy tales, Tolkien argues, there comes a moment near the end of the story, when things seem the most desperate. Suddenly, just as all hope is almost extinguished, salvation arrives in the form of a "sudden and miraculous grace: never to be counted on to recur" (Tolkien 1966, p. 86). This happy ending reflects a deeper truth about humanity, for, like the characters in such a story, we, too, were saved at the last moment by Christ's sacrifice on the cross, which redeemed us from the Fall of our first parents. Tolkien believed that the best mythopoeic works replicate on a smaller scale the salvific effects of the Gospels, the "good news" of Christ's death and resurrection. This is certainly what he tried to accomplish through his own creative endeavors. "Understanding eucatastrophe," writes Sam McBride (2020), "is crucial to grasping Tolkien and his work. As a structural element, it shapes many of his narratives" (p. 183). Indeed, several examples of eucatastrophe have been noted in *The Silmarillion*, *The Hobbit*, and *The Lord of the Rings*: it is evident, for instance, in Eärendil's success in convincing the Valar to return with him to assist his kinsmen in their long, losing war against Morgoth; in the sudden arrival of the eagles at the ends of the War of Wrath and the Battle of Five Armies; and in the destruction of the Ring of Power by Frodo on Mt. Doom. In addition, I would argue that the overthrow of dragons by individual humans also serves as a dramatic illustration of eucatastrophe in action, both in the medieval legends recounted by Jacobus de Voragine and in Tolkien's legendarium.

### 3.1. Eucatastrophe in The Golden Legend

Tolkien set down his thoughts on eucatastrophe long after Jacobus' death, of course, but uses the term himself when discussing older mythopoeic works in "On Fairy-stories." So, there is nothing anachronistic about asserting that some of the legends recounted by Jacobus could end in a eucatastrophic manner, as in the case of St. Donatus, who manages to kill a dragon either by striking it with a whip or spitting in its face—the point is that neither method could reasonably be expected to cause the death of such a powerful foe (de Voragine 1948, p. 434). Even female saints can overcome dragons with the help of the Lord. St. Margaret, for example, has only to cross herself in order to defeat the dragon that has swallowed her whole: "Or again, as another legend tells it, the monster seized her by the head and drew her into his maw, and it was then that she made the sign of the cross, and caused the dragon to burst, the damsel emerging unharmed from his body" (de Voragine 1948, p. 353). Several of the legends involving saints and dragons feature the taming of the latter. A few even manage to escape death. The pair of dragons confronted by St. Matthew fall asleep at his mere approach: "And when the populace gathered together, he commanded the dragons in the name of Jesus to go away, and they went off, harming no one" (de Voragine 1948, p. 562). This is eucatastrophe at its very best, for the ending of the story features the sudden and miraculous defeat of evil without unnecessary loss of life. It may be worth noting that Tolkien's own *Farmer Giles of Ham*, a "full-blown satire" of the

chivalric code and dragon-slaying stories like those recounted in *The Golden Legend*, also features a tame dragon who is subsequently let go (Reinhard 2020, p. 185).

Undoubtedly, the encounter that best exemplifies the link between eucatastrophe and dragons in *The Golden Legend* is that involving St. George, whose legend, after *Beowulf* and the *Völsunga saga*, is one of the most important medieval influences on Tolkien's overall conception of dragons (Lakowski 2015, p. 85). The dragon he must fight is an especially vicious one who has extracted from the people of Silene, Libya a daily tribute of two sheep. After that fails, they are forced to draw lots to determine which of them will be sacrificed to appease the dragon's hunger. The lot eventually falls to the king's daughter; despite her father's desperate pleas and offers of gold and other riches, the citizens of the town refuse to allow her to escape her fate. Just as she is being led away to her doom, George happens along, and, after learning the cause of her distress, he chivalrously agrees to serve as her deliverer. "My child, be without fear; for in the name of Christ I will succor thee!" (de Voragine 1948, p. 234). True to his word, he makes the sign of the cross with his sword before charging the dragon, whom he strikes down with his spear. George then instructs the maiden to tie her girdle around the beast's neck, after which she is able to lead the beast around like a pet. However, even a tame dragon is still a terrible thing to behold, and George promises to slay him if the people and their king will agree to be baptized as Christians. Jacobus concludes the affair with the happy report that "on that day twenty thousand men, and a multitude of women and children, received baptism" (de Voragine 1948, p. 234). In this story, eucatastrophe can be seen to work on many levels: the princess is saved by the sudden arrival of George, who, after making the sign of the cross, defeats a dragon in single-handed combat and then kills him, an event that leads to the unexpected but obviously fortuitous conversion of thousands of Muslims. Despite the inherent, postlapsarian weakness of George and all of these human saints, they all succeed in defeating one of the most powerful symbols of evil known to man. In these legends, one sees clear evidence of divine intervention in the everyday affairs of humans, or miracles, which Tolkien defined as "intrusions . . . into real or ordinary life" and regarded as one of the essential components in any story that ends in a eucatastrophe (Tolkien 2000, p. 100).

*3.2. Eucatastrophe and Tolkien's Dragons*

Perhaps ironically, his own aversion to making any explicit references to Christianity often obscures the frequency with which God or Eru intervenes in some of the key incidents in the history of Middle-earth. In an often-cited letter to Father Robert Murray, a close friend of the family, Tolkien describes *The Lord of the Rings* as "a fundamentally religious and Catholic work; unconsciously so at first, but consciously in the revision. That is why I have not put in, or have cut out, practically all references to anything like 'religion' . . . For the religious element is absorbed into the story and the symbolism" (Tolkien 2000, p. 172). Signs of this "religious element" abound and have been eagerly traced by many scholars over the years. However, I believe that it is evident, as well, in the deaths of Tolkien's dragons, which all depend on the same kind of divine intervention that he believed to be such an important factor in eucatastrophic stories. This can be seen, especially, in the final battles of the War of Wrath, after Eärendil sails to Valinor in an attempt to convince the Valar to leave their home and join with the peoples of Middle-earth in their fight against Morgoth. "The gods then move again, and great power comes out of the West, and the Stronghold of the Enemy is destroyed," Tolkien explains, drawing attention to the role of divine intervention in ending this decades-long conflict (Tolkien 2000, p. 150). However, victory is not achieved without one last rally by the forces of evil, featuring Morgoth's most dreadful creation: winged dragons. These are only defeated with the aid of Thorondor, king of eagles, and "all the great birds of heaven" (Tolkien 1977, p. 252), the arrival of whom Tolkien singles out for special attention as an example of eucatastrophe in *The Hobbit* (Tolkien 2000, p. 101). Perhaps divine intervention is the main difference between the glorious death of the dragons at the end of the War of Wrath, and the merely opportune stabbing of Glaurung by Túrin Turambar at Cared-en-Aras. Only the former

can be regarded as a genuine eucatastrophe: the latter, by contrast, does not lead to much, if any, happiness for either the characters or the reader.

The death of Smaug in *The Hobbit* shows how all of these ideas regarding postlapsarian weakness, eucatastrophe, and divine intervention can be syncretized in Tolkien's dragons. Over the years, Smaug has slaughtered countless elves, dwarves, and men, but the only individual challenger he faces in this book is a lowly hobbit—and not a very strong or adventurous one, at that. Though he is no one's idea of a hero, Bilbo Baggins somehow manages to not only survive his confrontation with the dragon but escape with vital information about the one flaw in Smaug's gem-encrusted armor, the "lone patch in the hollow of his left breast" (Tolkien [1937] 2001, p. 246). He reveals this weakness to the dwarves and is overheard by an old thrush while doing so: "a very old bird indeed," they tell him, "maybe the last left of the ancient breed that used to live about here," and reputed member of a "long-lived and magical race" (Tolkien [1937] 2001, p. 247). Perhaps this thrush is another one of Eru's winged servants on earth; in any event, he reports what he has learned to Bard, whose lucky bowshot with the famed Black arrow represents one of the clearest examples of eucatastrophe in the whole legendarium.

## 4. Conclusions: The Dragons of Modernity

Bard, a human, is only slightly more threatening to Smaug than a lone hobbit, but, if both are too weak to take on a dragon individually, they prove that the combined efforts of humans, hobbits, and animals are often strong enough to overcome any evil. This turns out to be only a kind of minor eucatastrophe, however, since the death of the dragon ends up provoking all-out war between elves, humans, and dwarves. As the first blows fall, the three races are suddenly forced to work together after being attacked by an army of goblins, which soon overwhelms them. Just when things are at their absolute bleakest, salvation appears in the form of Tolkien's favored ministers of Manwë and Eru. "The arrival of the eagles tips the balance of the battle from the goblins to the elves, dwarves, and men, but the setup for this ending really begins with the story, in the dwarves' lack of a plan or method for dealing with the dragon" (Northrup 2004, p. 833). Indeed, it is hard to imagine that the long chain of events leading from the dwarves' arrival at the Lonely Mountain to final victory at the Battle of Five Armies could have arisen solely out of chance. It is probably safer to say, as Jacobus and Tolkien believed, that a higher power was at work in guiding events to unfold the way they did. As the latter writes in a letter to his son Christopher:

> No man can estimate what is really happening at the present sub specie aeternitatis [Latin: "from an eternal perspective"]. All we do know, and that to a large extent by direct experience, is that evil labours with vast power and perpetual success—in vain; preparing always only the soil for unexpected good to sprout in. So it is in general, and so it is in our own lives . . . But there is still some hope that things may be better for us, even on the temporal place, in the mercy of God. And though we need all our natural human courage and guts (the vast sum of human courage and endurance is stupendous, isn't it?) and all our religious faith to face the evil that may befall us (as it befalls others, if God wills) still we may pray and hope. I do.
>
> (Tolkien 2000, p. 76)

Still today we "pray and hope" that that same power Tolkien appealed to for succor is still watching over us, ready to help us overcome the dragons of modernity.

### Possible Objections

It may be objected that dragons are far too rare to serve as indicators of postlapsarian weakness in Middle-earth, and that Tolkien does not employ them as representatives of evil. Furthermore, since things rarely improve after the death of his major dragons, it seems odd to cite them as illustrating how eucatastrophe works. Besides, is it not expected that dragons will be killed by their mortal foes? What is so miraculous about them dying, then, at the hands of a Bard or a Túrin? However, the scarcity of dragons is exactly the point; after

all, they are rare only after the First Age, since most of them are killed at the end of the War of Wrath. Before then, countless numbers are created by Morgoth—and it is worth noting that they *are* created by him, the primordial source of evil in Eä, or the universe, for use against Elves and the Edain, their long-lived and powerful human allies. Dragons must be rare during the Second and Third Ages because the human descendants of the Edain grew increasingly weaker and would not have been able to resist them in their former numbers. Although it is true that the death of a dragon does not always herald a happy ending for Tolkien's heroes (e.g., Glaurung), I believe that the deaths of Smaug and Ancalagon—killed in the War of Wrath—at least, deserve to be counted as eucatastrophes, since they both demonstrate the sudden turn from "sorrow and failure" towards joy that is an essential stage in every eucatastrophe (Tolkien 1966, p. 86). The dragon in *Beowulf* and Glaurung himself would seem to contradict the notion that dragons are always successfully defeated in battle, and that their deaths cannot therefore be unexpected. In closing, then, while I do not believe that his dragons are necessarily the *only* or even *best* means of illustrating these concepts in Tolkien's work, they can still be useful in doing so.

**Funding:** This research received no external funding.

**Institutional Review Board Statement:** Not applicable.

**Informed Consent Statement:** Not applicable.

**Data Availability Statement:** Not applicable.

**Conflicts of Interest:** The author declares no conflict of interest.

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
