# Peer review of "On the Symbolic Use of Dragons by Jacobus de Voragine and J. R. R. Tolkien"

_religions, doi:10.3390/rel14040552_

Round 1

Reviewer 1 Report

The article presents a great interest in dealing with a contemporary subject in clear consonance with the medieval past. The main hypothesis of the article is that Jacopo da Varazze's Legenda Aurea was used as a source by Tolkien in his conception of Dragons in order to write The Lord of the Rings.

However, this initial hypothesis is only supported by Judy Ann Ford's reference, which, when consulted, does not yield any hard data. With this in mind, the introduction to the article must provide much stronger arguments linking the two references. Otherwise, the article would turn into an exegesis of both texts looking for overlapping elements.

The medieval conception of the Dragon is influenced by multiple sources beyond the Legenda Aurea (the commentaries on the Book of the Apocalypse by Beatus of Liébana, the Bible itself, the physiologist, the martyrology...). The Legenda Aurea itself is a compilation of other legends of saints and there are many theories about its construction. With this in mind, there is a lack of further studies on the Legenda Aurea. In this respect, the author should be referred to throughout the text as Jacopo da Varazze, as Jacobus de Voragine is a widespread Latin corruption of the name, when the author's name in the original Italian language was otherwise. The article should deepen the study of this medieval source, as it is clear that the author has mastered Tolkien's work, and the work of the Legenda Aurea cannot become just an excuse to talk about the contemporary author.

A study is needed to compile the state of the art on medieval dragons, often combined with other animals such as the serpent, the amphisbena or the ketus. The dragon did not exist in the Middle Ages as a univocal animal: later translations of the Legenda Aurea and the Bible use the word Dragon to refer to a wide range of animals, very similar to each other, but which 20th century fantasy literature would understand as diverse creatures.

In some cases, the article presents the vision of the medieval dragon as being properly associated with Christianity, whereas it is classical Greco-Roman antiquity that constructs the medieval imaginary of this creature. It is worth introducing references to the classical legacy in the contribution of the medieval vision.

Some works linked to medieval iconography include interesting sources that the author should take into account, as the construction of the medieval image beyond the texts is also an important source for contemporary literature. Here are some references that should be consulted and included, especially those marked in bold:

Sharon Khalifa-Gueta (2022): «Mother of Dragons». Eikón / Imago 11, pp 79–92. <https://doi.org/10.5209/eiko.76756>.

Arroyo Cuadra, Sara (2012): «La iconografía del dragón y del grifo: mismo origen, distinto destino», Eikón Imago, vol. 1, n.° 1, pp. 105-118. <https://doi.org/10.5209/eiko.73244>.

Consiglieri, Nadia Mariana (2016): «Entre lo leonino, lo draconiano y lo humanoide. Notas sobre la representación pictórica de bestias y diablos en el área castellana y aragonesa (siglos xii-xiii)», Eikón Imago, vol. 5, n.° 2, pp. 69-106. <https://doi.org/10.5209/eiko.73495>.

Cirlot, Victoria (1987): «El dragón en la cultura medieval (Preámbulo a una exposición)», en L. Botey y V. Cirlot (publ.): El Drac en la cultura medieval. Exposició Fundació Caixa de Pensions. Catálogo de la exposición, Barcelona: Fundació Caixa de Pensions, pp. 20-25.

Ruiz-Domènec, José Enrique (1987): «La princesa y el dragón», en L. Botey y V. Cirlot (eds.): El Drac en la cultura medieval. Exposició Fundació Caixa de Pensions. Catálogo de la exposición, Barcelona: Fundació Caixa de Pensions, pp. 94-103.

Sadaune, Samuel (2016): Le fantastique au Moyen Âge, Rennes: Ouest-France-Edilarge.

Rodríguez Pérez, Diana (2006): «El combate contra la serpiente: el triunfo de la tierra velado bajo la aparente muerte del ofidio», De Arte, n.° 5, pp. 5-14. <http://dx.doi.org/10.18002/da.v0i5.1542>.

Ogden, Daniel (2013): Dragons, Serpents, and Slayers in the Classical and Early Christian Worlds. A Sourcebook, Oxford: Universidad de Oxford.

Ogden, Daniel (2013): Drakōn. Dragon Myth and Serpent Cult in the Greek and Roman Worlds, Oxford: Universidad de Oxford.

Malaxecheverría, Ignacio (1987): «El dragón en el bestiario medieval», en L. Botey y V. Cirlot (eds.): El Drac en la cultura medieval. Exposició Fundació Caixa de Pensions. Catálogo de la exposición, Barcelona: Fundació Caixa de Pensions, pp. 63-73.

Malaxecheverría, Ignacio (1991): «La lucha contra la regresión. El dragón-serpiente», en Fauna fantástica de la Península ibérica, San Sebastián: Kriselu, pp. 141-169.

Elvira Barba, Miguel Ángel (1994): «La iconografía del dragón en Bizancio», Erytheia: Revista de Estudios Bizantinos y Neogriegos, n.° 15, pp. 67-84.

Elvira Barba, Miguel Ángel (1997): «Los orígenes iconográficos del dragón medieval», en La tradición en la Antigüedad tardía, Antig. crist. XIV, Murcia, pp. 419-434.

Author Response

The article presents a great interest in dealing with a contemporary subject in clear consonance with the medieval past. The main hypothesis of the article is that Jacopo da Varazze's Legenda Aurea was used as a source by Tolkien in his conception of Dragons in order to write The Lord of the Rings.

However, this initial hypothesis is only supported by Judy Ann Ford's reference, which, when consulted, does not yield any hard data. With this in mind, the introduction to the article must provide much stronger arguments linking the two references. Otherwise, the article would turn into an exegesis of both texts looking for overlapping elements.

Thank you for the compliments. There isn’t much scholarship on the influence of the Golden Legend on Tolkien, so I have softened my claims about this somewhat and added a sentence to the intro clarifying my intentions in this paper. They are, more or less, to offer an “exegesis of both texts looking for overlapping elements” … I’m not really trying to argue that Jacobus’ text influenced Tolkien’s dragons so much as both authors, drawing from their shared Christian backgrounds, employ them in similar ways.

The medieval conception of the Dragon is influenced by multiple sources beyond the Legenda Aurea (the commentaries on the Book of the Apocalypse by Beatus of Liébana, the Bible itself, the physiologist, the martyrology...). The Legenda Aurea itself is a compilation of other legends of saints and there are many theories about its construction. With this in mind, there is a lack of further studies on the Legenda Aurea. In this respect, the author should be referred to throughout the text as Jacopo da Varazze, as Jacobus de Voragine is a widespread Latin corruption of the name, when the author's name in the original Italian language was otherwise. The article should deepen the study of this medieval source, as it is clear that the author has mastered Tolkien's work, and the work of the Legenda Aurea cannot become just an excuse to talk about the contemporary author.

I appreciate this suggestion, but the author is referred to as “Jacobus de Voragine” in (almost) every source I have consulted and is named as such on the cover of the edition of the Golden Legend I am using, as well. To avoid confusion, I would prefer to stick with this version of his name. I think this is how he is generally referred to in English (whatever that reveals about this language!).

A study is needed to compile the state of the art on medieval dragons, often combined with other animals such as the serpent, the amphisbena or the ketus. The dragon did not exist in the Middle Ages as a univocal animal: later translations of the Legenda Aurea and the Bible use the word Dragon to refer to a wide range of animals, very similar to each other, but which 20th century fantasy literature would understand as diverse creatures.

In some cases, the article presents the vision of the medieval dragon as being properly associated with Christianity, whereas it is classical Greco-Roman antiquity that constructs the medieval imaginary of this creature. It is worth introducing references to the classical legacy in the contribution of the medieval vision.

I have added a paragraph addressing the etymology of the word “dragon” and exploring some of the classical (Greek) traditions about them. I have also added a few notes about how the meaning of the word has changed over time. This paragraph has been combined with the one on “Dragons as symbols of evil,” since some of the Biblical discussion there seemed relevant to these points.

Some works linked to medieval iconography include interesting sources that the author should take into account, as the construction of the medieval image beyond the texts is also an important source for contemporary literature. Here are some references that should be consulted and included, especially those marked in bold:

I have added a paragraph on classical / medieval iconography. Time limitations precluded me from consulting all of the sources you provided, a list I greatly appreciate, but I did read through your recommendations in bold and incorporated them into the paper. Thank you very much for the suggestion to consider this topic. And, overall, I found this feedback incredibly helpful for improving the paper.

Reviewer 2 Report

The main arguments that a) the killing of a dragon is difficult to achieve due to a state of 'fall from grace' and b) must be therefore seen as a 'miracle' (and thus constitutes an example of a eucatastrophe) do not really convince. Dragons are far too rare to serve as indicators of a 'fall from grace' and, outside 'The Children of Turin', are rarely used as the representatives of evil. They do function, from a narratological point of view, as 'supreme obstacles' (a point Tolkien made about Beowulf), but not everything is well after their demise (a point also noticed by the author). So the second argument (the killing of the dragon constitutes a eucatastrophe) does not convince, either. The killing of Smaug by means of an arrow is rather 'modern' and, from a 'heroic' point of view, disappointing (see Shippey's discussion of this passage - Jackson had a hard time to make it into a heroic struggle ... ). A real 'eucatastrophic moment' would be, for example, the moment when the Rohirrim (surrounded by the enemies and getting ready for a last stand on the Pelennor Fields) realize that the Black Ships are not filled with corsairs, but with Aragorn and the Army of the Dead coming to their rescue. This is really a game-changer and unexpected for everybody (also for the first-time readers!). The killing of a dragon, however, is always expected, however hard it is, and does (in my view) not constitute a true example of eucatastrophe (unless handled in a way that makes the reader expect the hero to succumb and lose). The killing of Smaug could be construed (in hindsight) as a supreme example of Providence, making use of all the seemingly insignificant and unrelated past decisions of various people in the past to culminate in the one fatal shot. But seen from that point of view, it is not a sudden and 'unexpected' divine intervention.

Author Response

The main arguments that a) the killing of a dragon is difficult to achieve due to a state of 'fall from grace' and b) must be therefore seen as a 'miracle' (and thus constitutes an example of a eucatastrophe) do not really convince. Dragons are far too rare to serve as indicators of a 'fall from grace' and, outside 'The Children of Turin', are rarely used as the representatives of evil. They do function, from a narratological point of view, as 'supreme obstacles' (a point Tolkien made about Beowulf), but not everything is well after their demise (a point also noticed by the author). So the second argument (the killing of the dragon constitutes a eucatastrophe) does not convince, either. The killing of Smaug by means of an arrow is rather 'modern' and, from a 'heroic' point of view, disappointing (see Shippey's discussion of this passage - Jackson had a hard time to make it into a heroic struggle ... ). A real 'eucatastrophic moment' would be, for example, the moment when the Rohirrim (surrounded by the enemies and getting ready for a last stand on the Pelennor Fields) realize that the Black Ships are not filled with corsairs, but with Aragorn and the Army of the Dead coming to their rescue. This is really a game-changer and unexpected for everybody (also for the first-time readers!). The killing of a dragon, however, is always expected, however hard it is, and does (in my view) not constitute a true example of eucatastrophe (unless handled in a way that makes the reader expect the hero to succumb and lose). The killing of Smaug could be construed (in hindsight) as a supreme example of Providence, making use of all the seemingly insignificant and unrelated past decisions of various people in the past to culminate in the one fatal shot. But seen from that point of view, it is not a sudden and 'unexpected' divine intervention.

I have added a paragraph at the end of the paper responding to these points (or most of them) … sort of like a counter-argument. The scarcity of dragons in the Second and Third Ages supports my point about a postlapsarian weakness in the races of Middle-earth, I believe. Their creation by Morgoth would seem to suggest that it is safe to regard them as representatives of evil. Smaug’s death offers support for the idea of long planning on the part of Providence and a miraculous intervention at the last minute, guiding Bard’s arrow home … I don’t see why they have to be mutually exclusive, but understand that there are better examples of eucatastrophe in his work and valid criticisms to be made of the examples I have chosen to use. As I mention in closing, my intention here is not to argue that Tolkien’s dragons are the only or even the best way of illustrating these concepts … just that they can be helpful in doing so. Thank you for the suggestions.

Round 2

Reviewer 1 Report

The author has taken into account some of the considerations raised in the previous review.

Reviewer 2 Report

no further corrections necessary